# Reemergence of high-$T_c$ superconductivity in the (Li$_{1-x}$Fe$_x$)OHFe$_{1-y}$Se under high pressure

J.P. Sun[1,2], P. Shahi[1,2], H.X. Zhou[1,2], Y.L. Huang[1,2], K.Y. Chen[1,2], B.S. Wang[1,2], S.L. Ni[1,2], N.N. Li[3], K. Zhang[3], W.G. Yang[3,4], Y. Uwatoko[5], G. Xing[6,7], J. Sun[6], D.J. Singh[6], K. Jin[1,2], F. Zhou[1,2], G.M. Zhang[8], X.L. Dong[1,2], Z.X. Zhao[1,2] & J.-G. Cheng[1,2]

In order to elucidate pressure-induced second superconducting phase (SC-II) in A$_x$Fe$_{2-y}$Se$_2$ (A = K, Rb, Cs, and Tl) having an intrinsic phase separation, we perform a detailed high-pressure magnetotransport study on the isoelectronic, phase-pure (Li$_{1-x}$Fe$_x$)OHFe$_{1-y}$Se single crystals. Here we show that its ambient-pressure superconducting phase (SC-I) with a critical temperature $T_c \approx 40$ K is suppressed gradually to below 2 K and an SC-II phase emerges above $P_c \approx 5$ GPa with $T_c$ increasing progressively to above 50 K up to 12.5 GPa. Our high-precision resistivity data uncover a sharp transition of the normal state from Fermi liquid for SC-I to non-Fermi liquid for SC-II phase. In addition, the reemergence of high-$T_c$ SC-II is found to accompany with a concurrent enhancement of electron carrier density. Without structural transition below 10 GPa, the observed SC-II with enhanced carrier density should be ascribed to an electronic origin presumably associated with pressure-induced Fermi surface reconstruction.

[1] Beijing National Laboratory for Condensed Matter Physics and Institute of Physics, Chinese Academy of Sciences, Beijing, 100190, China. [2] School of Physical Sciences, University of Chinese Academy of Sciences, Beijing, 100190, China. [3] Center for High Pressure Science and Technology Advanced Research (HPSTAR), Shanghai, 201203, China. [4] High Pressure Synergetic Consortium (HPSynC), Geophysical Laboratory, Carnegie Institution of Washington, 9700 S Cass Avenue, Argonne, IL 60439, USA. [5] The Institute for Solid State Physics, University of Tokyo, Kashiwa, Chiba 277-8581, Japan. [6] Department of Physics and Astronomy, University of Missouri, Columbia, MO 65211-7010, USA. [7] College of Materials Science and Engineering and Key Laboratory of Automobile Materials of MOE, Jilin University, Changchun, 130012, China. [8] State Key Laboratory of Low Dimensional Quantum Physics and Department of Physics, Tsinghua University, Beijing, 100084, China. J. P. Sun, P. Shahi, and H. X. Zhou contributed equally to this work. Correspondence and requests for materials should be addressed to X.L.D. (email: dong@iphy.ac.cn) or to J.-G.C. (email: jgcheng@iphy.ac.cn)

Among the iron-based superconductors, the structural simplest FeSe and its derived materials have attracted tremendous attention recently due to its peculiar electronic properties and the great tunability of the superconducting transition temperature $T_c$. The bulk FeSe displays a relatively low $T_c \approx 8.5$ K within the peculiar nonmagnetic nematic phase below $T_s \approx 90$ K[1]. By intercalating some alkali-metal ions, ammonia, or organic molecules in between the adjacent FeSe layers, such as in $A_xFe_{2-y}Se_2$ (A = K, Rb, Cs, and Tl)[2,3], $A_x(NH_3)_yFeSe$[4], and (Li,Fe)OHFeSe[5,6], high-$T_c$ superconductivity with $T_c$ above 30–40 K has been successfully achieved. More surprisingly, when a single unit-cell FeSe film is fabricated on the SrTiO$_3$ substrate, its $T_c$ can be raised up to 65–100 K[7,8]. Here we refer these high-$T_c$ superconductors derived directly from FeSe as the SC-I phase. The superconducting mechanism for these SC-I phases has been subjected to extensive investigations, and the observed common Fermi surface (FS) topology consisting of only electron pocket in the Brillouin zone corners suggests that the electron doping plays an essential role for achieving high $T_c$[9–11], in agreement with the gate-voltage regulation experiments on the FeSe flakes[12].

Starting from the SC-I phase in $A_xFe_{2-y}Se_2$, Sun et al.[13] had reported a pressure-induced sudden reemergence of a second superconducting phase (denoted as SC-II hereafter) with higher $T_c$ up to 48.7 K above ~10 GPa. A similar SC-II phase has also been observed in $Cs_{0.4}(NH_3)_yFeSe$ under high pressure[14]. Although the reemergence of SC-II phase with higher $T_c$ is quite intriguing and different pairing symmetry has been proposed theoretically[15], the intrinsic superconducting- and normal-state properties have been poorly characterized so far due to some sample and technical difficulties. For example, $A_xFe_{2-y}Se_2$ superconductors are prone to phase separation accompanied with the intergrowth of antiferromagnetic insulating $A_2Fe_4Se_5$ phase[16]. In addition, only polycrystalline samples have been studied under pressure for $Cs_{0.4}(NH_3)_yFeSe$, which is extremely sensitive to air[14]. Moreover, high-pressure technique capable of both large pressure capacity and good hydrostaticity is required

in order to obtain reliable superconducting- and normal-state properties. Therefore, these complexities have hampered a proper understanding on the intriguing SC-II phase of these FeSe-derived systems.

In order to approach this intriguing problem, we turn our attention to the recently discovered $(Li_{1-x}Fe_x)OHFe_{1-y}Se$[5,6], which is free from phase separation, relatively stable in air, and more importantly, can be obtained in high-quality single crystals via a specially designed hydrothermal ion-exchange method[17]. $(Li_{0.84}Fe_{0.16})OHFeSe$ with an optimal $T_c \approx 41$ K is heavily electron doped having only electron pockets at the Brillouin zone corners, similar as $A_xFe_{2-y}Se_2$ and monolayer FeSe/SrTiO$_3$ film[10,18]. In addition, the distance between two adjacent FeSe layers in $(Li_{1-x}Fe_x)OHFe_{1-y}Se$ is much larger than that in bulk FeSe and $A_xFe_{2-y}Se_2$, which signals a weak interlayer interaction and an enhanced two-dimensional nature of the electronic structure[19]. It thus has been considered as a better proxy of the monolayer FeSe film but is more stable and free from interface effects[10]. These factors together make it indispensable to perform a high-pressure study on $(Li_{1-x}Fe_x)OHFe_{1-y}Se$ single crystals.

Here we report detailed magnetotransport measurements on the $(Li_{1-x}Fe_x)OHFe_{1-y}Se$ single crystals under hydrostatic pressures up to 12.5 GPa with a cubic anvil cell (CAC) apparatus[20]. We find that the ambient-pressure SC-I phase is suppressed gradually with increasing pressure to $P_c \approx 5$ GPa, above which a new SC-II phase with higher $T_c$ over 50 K emerges gradually. Importantly, our high-precision resistivity data enable us to uncover a sharp transition of the normal state from Fermi liquid for SC-I to non-Fermi liquid for SC-II phase. In addition, the reemergence of higher $T_c$ SC-II phase is found to accompany with a concurrent enhancement of electron carrier density. Such information was unavailable in previous high-pressure studies on the FeSe-derived superconductors. The present work thus provides positive correlations between the high-$T_c$ superconductivity in SC-II with a FS reconstruction, which is not induced by a structural transition as confirmed by our high-pressure structural study.

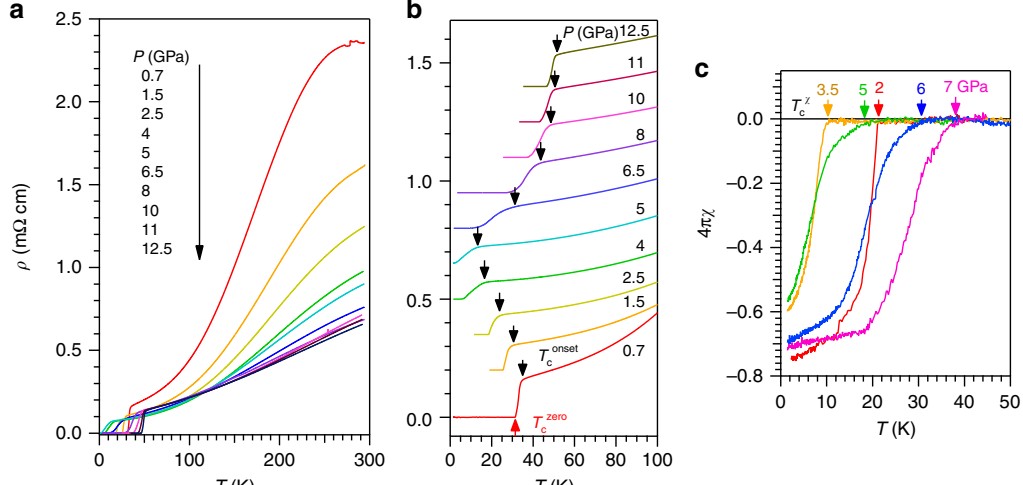

**Fig. 1** High-pressure resistivity and AC magnetic susceptibility for $(Li_{1-x}Fe_x)OHFe_{1-y}Se$. **a** $\rho(T)$ curves in the whole temperature range illustrating the overall behaviors under pressure up to 12.5 GPa. **b** $\rho(T)$ curves below 100 K illustrating the variation with pressure of the superconducting transition temperatures. Except for data at 0.7 GPa, all other curves in **b** have been vertically shifted for clarity. The onset $T_c^{onset}$ (down-pointing arrow) was determined as the temperature where resistivity starts to deviate from the extrapolated normal-state behavior, while the $T_c^{zero}$ (up-pointing arrow) was determined as the zero-resistivity temperature. **c** $4\pi\chi(T)$ curves measured under different pressures up to 7 GPa. The superconducting diamagnetic signal appears below $T_c^{\chi}$

## Results

**High-pressure resistivity.** Figure 1a shows the temperature dependence of resistivity $\rho(T)$ for a $(Li_{1-x}Fe_x)OHFe_{1-y}Se$ single crystal ($x \approx 0.16$, $y \approx 0.02$, and $T_c \approx 40$ K at ambient pressure) measured under various hydrostatic pressures up to 12.5 GPa in the whole temperature range. As can be seen, $\rho(T)$ in the normal state first decreases significantly and then becomes nearly unchanged above 6.5 GPa; the broad hump feature at high temperature also smears out gradually upon increasing pressure. The superconducting $T_c$ displays a non-monotonic variation with pressure, which can be seen more clearly from the vertically shifted $\rho(T)$ data below 100 K as shown in Fig. 1b. Here we define the onset $T_c^{onset}$ (down-pointing arrow) as the temperature where $\rho(T)$ starts to deviate from the extrapolated normal-state behavior, and determine $T_c^{zero}$ (up-pointing arrow) as the zero-resistivity temperature. As can be seen, upon increasing pressure to 5 GPa, $T_c^{onset}$ is suppressed gradually to ~13 K and $T_c^{zero}$ can hardly be defined down to 1.4 K, the lowest temperature in the present study. Interestingly, when increasing pressure to 6.5 GPa, a broad superconducting transition appears again with the $T_c^{onset}$ raised to ~31 K and $T_c^{zero}$ at ~12 K, thus evidencing the emergence of the SC-II phase. With further increasing pressure, both $T_c^{onset}$ and $T_c^{zero}$ move up progressively and the superconducting transition becomes sharper. Finally, $T_c^{onset}$ and $T_c^{zero}$ reach 52.7 and 46.2 K, respectively, at $P_{max} = 12.5$ GPa. A closer inspection of the $\rho(T)$ data in Fig. 1b also reveals a gradual evolution of the temperature dependence of normal-state resistivity under pressure, which will be discussed in detailed below.

**AC magnetic susceptibility.** The superconducting transitions have been further verified by the AC magnetic susceptibility $4\pi\chi$ $(T)$ shown in Fig. 1c, in which the superconducting diamagnetic signal appears below $T_c^\chi$ as indicated by the arrows. The obtained $T_c^\chi$ first decreases with pressure, reverses the trend near $P_c \approx 5$ GPa, and then increases quickly with further increasing pressure, in well agreement with the resistivity data. In addition, the transition in $4\pi\chi(T)$ is broad when the resistivity transition is broad for $5 < P < 8$ GPa. Nevertheless, the superconducting shielding volume reaching over 60–70% confirmed the bulk nature of the observed superconductivity in both SC-I and SC-II phases.

**Temperature-pressure phase diagram.** The pressure dependences of the obtained $T_c^{onset}$, $T_c^{zero}$, and $T_c^\chi$ for the studied $(Li_{1-x}Fe_x)OHFe_{1-y}Se$ are displayed in Fig. 2a, which evidenced explicitly the gradual suppression of the SC-I phase followed by the reemergence of the SC-II phase above $P_c \approx 5$ GPa. Such an evolution of superconducting phases is clearly different from that of bulk FeSe under high pressure[21,22]. It looks that the SC-II phase will exhibit a dome-shaped $T_c(P)$ with the maximum taking place around 12–13 GPa. It is interesting to note that in the SC-I region $T_c^\chi$ agrees well with $T_c^{zero}$ as commonly seen in most superconductors, whereas in the SC-II region $T_c^\chi$ follows the $T_c^{onset}$, implying that a considerable superconducting volume already appears near $T_c^{onset}$ despite of a broad transition. Although the observation of pressure-induced SC-II phase in $(Li_{1-x}Fe_x)OHFe_{1-y}Se$ in the present study is qualitatively similar with those reported in $A_xFe_{2-y}Se_2$ and $Cs_{0.4}(NH_3)_yFeSe$[13,14], there are some quantitative differences in comparison with those previously studies: (i) the obtained $T_c^{onset}$ here is higher, exceeding 50 K for the first time; (ii) $T_c^{zero}$ that has never been achieved for the SC-II phase in the previous studies using the diamond anvil cell (DAC) is successfully reached here due to a better sample quality and improved hydrostaticity in the CAC; (iii) the SC-II

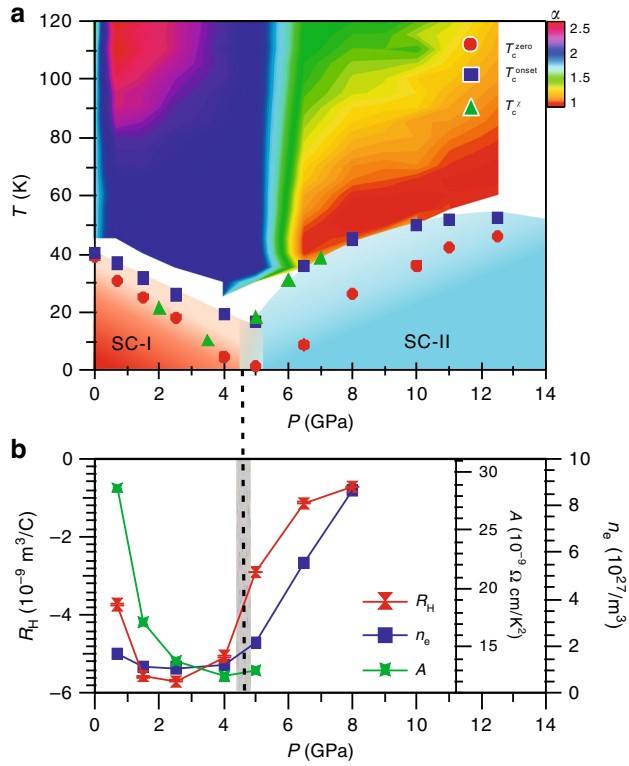

**Fig. 2** $T$–$P$ phase diagram of $(Li_{1-x}Fe_x)OHFe_{1-y}Se$ with an ambient-pressure $T_c = 40$ K. **a** Pressure dependence of the superconducting transition temperatures $T_c$s and a contour color plot of the normal-state resistivity exponent $\alpha$ up to 12.5 GPa. The values of $T_c^{onset}$, $T_c^{zero}$, and $T_c^\chi$ were determined from the high-pressure resistivity and AC magnetic susceptibility. The temperature dependence of $\alpha$ are extracted from $d\ln(\rho - \rho_0)/d\ln T$ for each pressure. **b** Pressure dependences of the Hall coefficient $R_H$ and the electron density $n_e$ up to 8 GPa extracted from the transverse resistivity $\rho_{xy}$. Pressure dependence of the resistivity coefficient $A$ in the plot of $\rho \sim AT^2$ below 5 GPa. A double-domed $T_c(P)$ accompanied with distinct normal-state properties for each superconducting phase is clearly observed. The reemergence of the SC-II phase with higher $T_c$ is accompanied with a dramatic enhancement of the carrier density $n_e$

phase appears gradually and exists in a wide pressure range. We have measured another $(Li_{1-x}Fe_x)OHFe_{1-y}Se$ sample with a lower $T_c \approx 28$ K at ambient pressure and observed very similar behaviors featured by two superconducting domes separated at a lower critical pressure of $P_c \approx 3$ GPa. Details can be found in the Supplementary Fig. 1. These experiments thus confirm that the pressure-induced reemergence of SC-II phase is likely a universal phenomenon in the $(Li_{-1-x}Fe_x)OHFe_{1-y}Se$ system, or even in the FeSe-derived high-$T_c$ superconductors taking together the previous studies [13,14].

**Transition from Fermi liquid to non-Fermi liquid around $P_c$.** To uncover the origin of such an intriguing phenomenon, experimentally we need to first characterize the normal-state properties, which are usually correlated tightly with the superconducting states for unconventional superconductors. A distinct change on the temperature dependence of normal-state $\rho(T)$ has already been noticed in Fig. 1b. To quantify this evolution, we display the $\rho(T)$ data in a double-logarithmic plot of $\log(\rho - \rho_0)$ vs. $\log T$ in Fig. 3a, where $\rho_0$ is the residual resistivity at zero temperature. The slope of these curves corresponds to the resistivity exponent $\alpha$ in $\rho \propto T^\alpha$, which

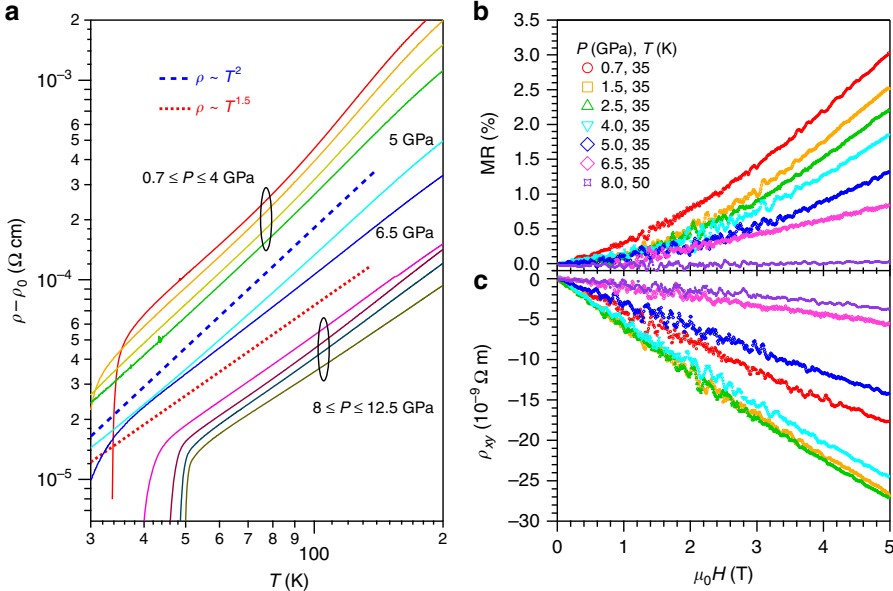

**Fig. 3** Normal-state transport properties of $(Li_{1-x}Fe_x)OHFe_{1-y}Se$ under high pressure. **a** A double-logarithmic plot of $(\rho - \rho_0)$ vs. $T$ illustrating the variation with pressure of the normal-state resistivity from the Fermi-liquid $\rho \sim T^2$ for $P < 5$ GPa to non-Fermi-liquid $\rho \sim T^{1.5}$ behavior for $P > 6.5$ GPa. Except for the curve at 5 GPa, all other curves have been vertically shifted for clarity. Field dependence of **b** the magnetoresistance MR and **c** the transverse resistivity $\rho_{xy}$ at the normal state just above $T_c$ under various pressures. The Hall coefficient $R_H$ is determined from the field derivative of $\rho_{xy}$, $R_H \equiv d\rho_{xy}/dH$, at each pressure

evolves from a Fermi-liquid $\alpha = 2$ for $0.7 \leq P \leq 4$ GPa, through some intermediate $1.5 < \alpha < 2$ for $P = 5$ and 6.5 GPa, and finally to non-Fermi-liquid $\alpha \leq 1.5$ for $P > 6.5$ GPa. Such an evolution can be visualized more profoundly in a contour plot of the resistivity exponent $\alpha \equiv d\log(\rho - \rho_0)/d\log T$ superimposed in Fig. 2a. The observed sharp transition of normal-state behavior thus signals distinct superconducting states for the SC-I and SC-II phases. In particular, the nearly linear-in-$T$ behavior for the SC-II phase resembles those of the optimal doped cuprates and iron-pnictides superconductors, thus implying an unconventional mechanism for the emergent SC-II phase[23]. We want to underline that our high-precision resistivity data enable us to unveil the non-Fermi-liquid normal state of the SC-II phase for the first time.

**Enhanced carrier density above $P_c$.** In order to gain further insights into the peculiar non-Fermi-liquid behavior of SC-II phase, we tried to probe the electronic structure information via measurements of magnetoresistance (MR) and Hall effect under pressure. Figure 3b, c displays the field dependence of in-plane $MR(H) \equiv [\rho(H)/\rho(0) - 1] \times 100\%$ and Hall resistivity $\rho_{xy}(H)$ in the normal state just above $T_c$ under various pressures up to 8 GPa. As can be seen, the MR is small and decreases gradually from 3% at 0.7 GPa to below 0.5% at 8 GPa. All $\rho_{xy}(H)$ curves exhibit a linear-in-$H$ behavior with a negative slope, signaling that the electron-type carriers dominate the charge transport in both the SC-I and SC-II phases. This observation also distinguishes the SC-II phase of $(Li_{1-x}Fe_x)OHFe_{1-y}Se$ from the high-$T_c$ phase of FeSe under high pressure showing the hole-dominated charge transport. In contrast with the monotonic decrease of MR, $\rho_{xy}$ displays a non-monotonic variation with pressure. Here we obtained the Hall coefficient $R_H \equiv d\rho_{xy}/dH$ as the slope of a linear fitting to $\rho_{xy}(H)$, and plotted the pressure dependence of $R_H(P)$ in Fig. 2b. As can be seen, $R_H$ is negative, and its magnitude first increases slightly with pressure and then experiences a quick reduction above 4 GPa. Assuming a simple

one-band contribution, the electron-type carrier density can be estimated as $n_e = -1/(R_H \cdot e)$. As shown in Fig. 2b, $n_e$ takes a relatively constant value of $\sim 2 \times 10^{27}$ m$^{-3}$ within the SC-I region for $P < 5$ GPa, above which it increases linearly to a large value of $\sim 9 \times 10^{27}$ m$^{-3}$ at 8 GPa, tracking nicely the trend of $T_c(P)$. These results demonstrate that the emergence of SC-II phase with higher $T_c$ is accompanied with a concurrent enhancement of electron carrier density. Such a positive correlation between $T_c$ and $n_e$ is consistent with the observations in the FeSe-based superconductors as mentioned above, but the origin of the pressure-induced enhancement of $n_e$ in the SC-II phase deserves in-depth investigations.

**High-pressure synchrotron X-ray diffraction.** To this end, we first checked if a structural transition takes place near $P_c \approx 5$ GPa. Figure 4a displays the high-pressure synchrotron X-ray diffraction (SXRD) patterns of $(Li_{1-x}Fe_x)OHFe_{1-y}Se$ measured at room temperature up to 14 GPa. All the peaks can be indexed in the tetragonal $P4/nmm$ (No. 129) space group plus a trace amount of Selenium (Se) secondary phase (space group $P3_121$) with the main peak located near $\sim 12°$. As can be seen, no obvious structural transition can be discerned in the investigated pressure range. The relative peak intensities are altered when applying pressure above 0.8 GPa due to the development of preferred orientation, as exemplified by the (200) peak near 20°. In addition to the preferred orientation, the presence of light elements H, O, and Li, and the significant peak broadening has hampered reliable Rietveld structural refinements on these SXRD data. We thus have applied the LeBail fit to the SXRD patterns and extracted the unit-cell parameters as a function of pressure as depicted in Fig. 4b–d. As can be seen, both the lattice parameters $a$, $c$, and the unit-cell volume $V$ decrease smoothly up to 10 GPa, above which $a$ and $c$ experiences some abnormal variations. Given a larger compressibility of the $c$ axis, the $c/a$ ratio decreases monotonically at least up to 10 GPa, Fig. 4c, which cannot explain the non-monotonic variations of $T_c(P)$

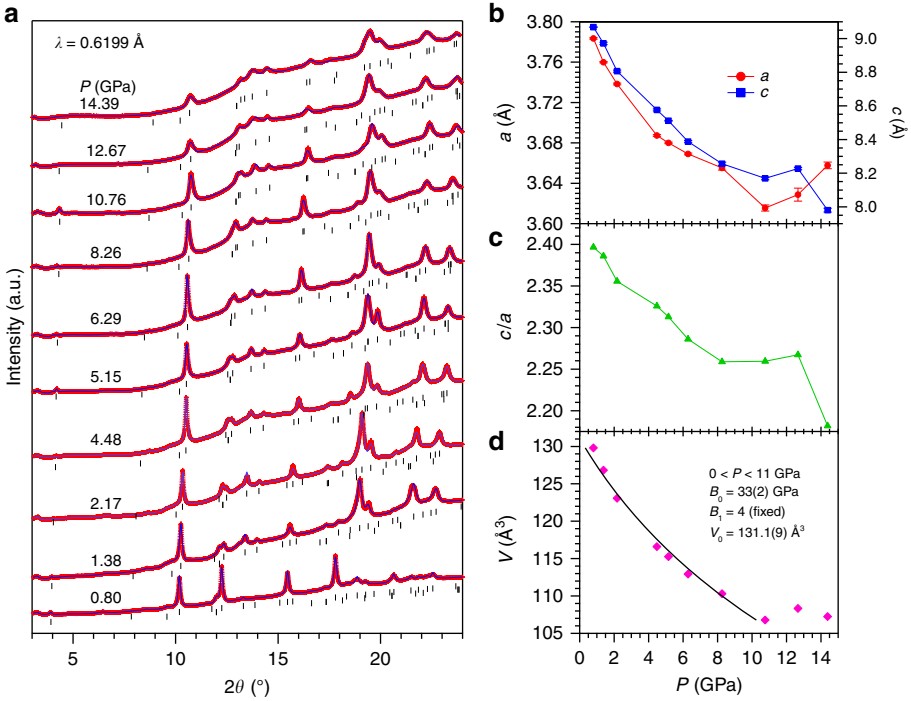

**Fig. 4** High-pressure synchrotron X-ray diffraction. **a** SXRD patterns of $(Li_{1-x}Fe_x)OHFe_{1-y}Se$ fitted with LeBail method. The first and second rows of tick marks in **a** represent the Bragg positions of the tetragonal $P4/nmm$ phase of $(Li_{1-x}Fe_x)OHFe_{1-y}Se$ and the secondary phase Se with space group $P3_121$. **b–d** Pressure dependence of unit-cell parameters $a$, $c$, $V$, and the $c/a$ ratio. The solid line in **d** is the Birch–Murnaghan fitting curve used to extract the bulk modulus given in the inset

shown in Fig. 2a[24]. Since the SXRD peaks become relatively broad above 10 GPa due to the solidification of liquid pressure transmitting medium, the anomalous structure changes above 10 GPa deserve further studies with better resolved SXRD patterns by employing the gas pressure transmitting medium. But, our present high-pressure structural study rules out any structural transition below 10 GPa as the possible cause for the observed enhancement of carrier density and the emergence of SC-II phase.

## Discussion

Without a structural transition taking place at $P_c \approx 5$ GPa, we are left with an electronic origin for the observed SC-II phase. Unfortunately, a direct experimental probe of the electronic structure near FS, e.g., with angle-resolved photoemission spectroscopy (ARPES), is impossible under high pressure. We then resorted to first-principles calculations as a function of pressure to check changes in the electronic structure. As detailed in the Supplementary Note 1, however, the calculated electronic structures up to 10 GPa do not show obvious changes related to our experimental findings here. Such a failed effort might arise from the fact that the band structural calculations cannot properly reproduce the experimentally observed FSs in (Li,Fe)OHFeSe via ARPES[10]. We need more dedicated calculations to address this issue in the future. Below we discuss briefly some possibilities that could lead to a FS reconstruction under pressure.

Recently, a second high-$T_c$ dome and a second enhancement of superconductivity have been reported in the heavily K-deposed FeSe film grown on SiC[25] and the $SrTiO_3$ substrates[26], respectively. By taking advantage of the in-situ ARPES measurements, the second enhancement of superconductivity in the latter case has been attributed to a Lifshitz transition associated with the emergence of an electron pocket at the Γ point of Brillouin zone

center[26]. Similarly, a second superconducting dome was also reported very recently in the surface K-dosed $(Li_{0.8}Fe_{0.2})OHF$-eSe[27], and was ascribed to emergent electron pocket at the Γ point. But the obtained $T_c$ is much lower than that of SC-II observed here under high pressure. Since no extra electron carriers were purposely doped into $(Li_{1-x}Fe_x)OHFe_{1-y}Se$ in the present case, the dramatic enhancement of carrier density $n_e$ and $T_c$ above 5 GPa cannot be ascribed to a doping-induced Lifshitz transition. Some other factors might play a role at high pressure. On the one hand, the magnetism of the (Li,Fe)OH layer could be suppressed by pressure, releasing some charge carriers into the FeSe layer. However, such a scenario is unlikely since the reemergence of SC-II phase has been observed universally in different classes of FeSe-derived systems. Alternatively, a Lifshitz transition in the heavily electron-doped FeSe layers might take place via a pressure-induced FS reconstruction. Recent scanning tunneling spectroscopy study on the $(Li_{1-x}Fe_x)OHFeSe$ single crystal has identified two electron pockets at the M point associated with the $d_{xy}$ and $d_{xz}/d_{yz}$ orbitals, respectively[28]. The observed $(\pi, 0.67\pi)$ wave vector in the spin resonance spectroscopy with inelastic neutron scattering is consistent with the nesting vector between the two-dimensional electron Fermi pockets[29]. Whether these electron pockets at M point undergo reconstruction or another electron/hole pockets emerge near Γ point deserve further theoretical studies. According to a recent ARPES study on FeSe films by Phan et al.[30], a compression strain realized in $FeSe/CaF_2$ will enlarge significantly both the hole and electron FSs in comparison with the strain-free FeSe. It is thus possible that compression on the FeSe planes above the critical pressure $P_c$ can result in a FS reconstruction leading to a larger FS volume. In any case, the reemergence of higher $T_c$ SC-II phase developed from the unusual non-Fermi-liquid normal state with enhanced electronic carrier density outlines important constrains for further investigations.

Finally, it is noteworthy that the normal-state resistivity of the cuprate superconductors, e.g., the overdoped $La_{2-x}Sr_xCuO_4$ and $La_{2-x}Ce_xCuO_4$[31,32], behaves as $\rho(T) \sim T^{1.6}$ at the verge of the superconducting dome, which has been attributed to quantum criticality. The observation of similar power-law behavior near the border of SC-II dome in the present $(Li_{1-x}Fe_x)OHFe_{1-y}Se$ thus points to the common physics that awaits for in-depth explorations in future.

In summary, we have measured the resistivity of $(Li_{1-x}Fe_x)OHFe_{1-y}Se$ single crystal under hydrostatic pressures up to 12.5 GPa with a CAC apparatus, and observed a gradual suppression of superconductivity followed by reemergence of a high-$T_c$ SC-II phase above $P_c \approx 5$ GPa. The highest $T_c$ reaches ~52 K, which is the highest among the bulk form of FeSe-derived superconductors. The SC-II phase is confirmed to develop from a peculiar non-Fermi-liquid normal state featured by dominant electron-type charge carriers and enhanced carrier density. Since no any structural transition was detected below 10 GPa, the observed SC-II phase with enhanced carrier density should be ascribed to an electronic origin associated with FS remonstration.

## Methods

**Sample preparation.** $(Li_{1-x}Fe_x)OHFe_{1-y}Se$ single crystals used in the present study were grown with a hydrothermal ion-exchange technique by using a large insulating $K_{0.8}Fe_{1.6}Se_2$ crystal as a matrix. Details about the crystal growth and sample characterizations at ambient pressure can be found in the previous study[17].

**High-pressure resistivity and AC magnetic susceptibility.** High-pressure transport and AC magnetic susceptibility were performed in the palm CAC apparatus[20]. The standard four-probe method was employed for resistivity measurement with the current applied within the $ab$ plane and the magnetic field along the $c$ axis. The $\rho_{xy}(H)$ and $\rho_{xx}(H)$ data were anti-symmetrized (symmetrized) with respect to the magnetic field between +5 and −5 T. Glycerol was employed as the pressure transmitting medium. The pressure values inside the CAC were calibrated at room temperature by observing the characteristic transitions of bismuth. The mutual induction method was used for the AC magnetic susceptibility measurements.

**High-pressure SXRD.** High-pressure SXRD was measured with DAC at the BL15U1 beamline, Shanghai Synchrotron Radiation Facility of China. Glycerol was used as the pressure medium. The pressure in DAC was monitored with the ruby fluorescence method.

**Data availability.** The data that support the findings of this study are available from the corresponding authors upon reasonable request.

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

## Acknowledgements

This work is supported by the National Science Foundation of China (Grant Nos. 11574377, 11574370, and U1530402), the National Basic Research Program of China (Grant Nos. 2014CB921500, 2017YFA0303000, and 2016YFA0300301), and the Strategic Priority Research Program and Key Research Program of Frontier Sciences of the Chinese Academy of Sciences (Grant Nos. XDB07020100, QYZDB-SSW-SLH013, QYZDY-SSW-SLH001, and QYZDY-SSW-SLH008). Y.U. is supported by the JSPS KAKENHI (Grant No. 15H03681). The authors would like to thank Dr. Aiguo Li and Dr. Ke Yang for technical support at the BL15U1 beamline, Shanghai Synchrotron Radiation Facility of China.

## Author contributions

J.-G.C. and X.L.D. conceived the project. H.X.Z., Y.L.H., S.L.N., K.J., F.Z., X.L.D. and Z.X.Z. synthesized the (Li,Fe)OHFeSe single crystals. J.P.S., P.S., K.Y.C., B.S.W., Y.U. and J.-G.C. performed the high-pressure resistivity and AC magnetic susceptibility measurements with the cubic anvil cell apparatus. J.P.S., N.N.L., K.Z. and W.G.Y. measured

high-pressure SXRD. G.X., J.S., D.J.S., and G.M.Z. performed theoretical calculations and analyses on the electronic structures under pressure. All authors discussed the results. J.P.S. and J.-G.C. wrote the paper with inputs from all authors.

## Additional information

**Competing interests:** The authors declare no competing financial interests.

