## [Peer Review File · Nature Communications]

Reviewer #1 (Remarks to the Author):

This manuscript reports the magnetotransport measurements under high pressure up to 12.5 GPa the single crystal $(\text{Li}_{1-x}\text{Fe}_x)\text{OHFe}_{1-y}\text{Se}$, which is considered to be free from phase separation and relatively stable in air. The authors found that the ambient-pressure SC-I phase is suppressed with increasing pressure to $P_c \sim 6$ GPa, above which a new SC-II phase emerges with T_c increasing with pressure to achieve T_c over 50 K. The reemergence of higher T_c SC-II phase was found to accompany with a concurrent enhancement of electron carrier density. In addition, the authors noted based on the normal state resistivity data a transition of from a Fermi liquid for SC-I phase to a non-Fermi-liquid for SC-II phase. The authors concluded that this work provides positive correlations between the high- T_c superconductivity in SC-II with a Fermi surface reconstruction, which is not induced by a structural transition as confirmed by our high-pressure structural study. This is indeed an interesting new result that is suitable for publication on Nature Communication. However, there are questions the authors need to address before the paper can be accepted for publication:

1. The x-ray results—need to show the refinement results to confirm no structural change above 6.29 GPa; the refined lattice parameters that show little changes above 8 GPa, seem not consistent with the XRD patterns.
2. The appearance of the SC-II above 6 GPa and reaches $T_c \sim 40$ K at ~ 8 GPa is similar to that observed in FeSe under pressure (See Medvedev, et al., DOI:10.1038/NMAT2491). The authors need to provide clear statement to make the distinction between the two.

Reviewer #2 (Remarks to the Author):

The paper by J.-G. Cheng on the reemergence of superconductivity of $(\text{Li}_{1-x}\text{Fe}_x\text{OH})\text{FeSe}$ is quite novel and important for the superconducting community. The conclusions are original and most of the measurements are convincing with respect to finding a second superconducting dome with an even higher T_c in bulk samples. I do not doubt the resistivity or magnetization measurements that ultimately build the phase diagram of Figure 1, but I do have concerns in identifying the phase responsible for this second dome.

I suggest the following changes before publication.

1.) The authors state that Rietveld refinements were performed to extract the lattice parameters that are plotted in Figure 4b,c,d. However, the curves in Figure 4a are not fit to anything. I could not find any in the Supplemental Information file either. Somehow, I do not believe that full structural Rietveld fits were performed because the powder patterns do not appear to be of high enough quality to extract lattice parameters, structural parameters (not every atom is in a special position), and other powder parameters (such as background and profile shape). I would suggest that the authors perform, instead, simple LeBail fits and present it as such for the system of interest. This is sufficient for lattice parameters, anyway, and preferred orientation is not even a problem with this type of analysis.

2.) Not every peak is indexed by the $P4/nmm$ structure in the powder patterns. There is one labeled by an asterisk. What is it? Also what are the extra phases in the pattern? This could be important for the phase that is responsible for the high T_c of 52 K. We are assuming that it is $(\text{Li}_{1-x}\text{Fe}_x\text{OH})\text{FeSe}$, but it might not be. Perhaps it is a new body-centered phase where a collapse of the c -parameter occurs (as in some 122-iron arsenides) that would lead to a complete rearrangement of the electronic structure. Note, that I am not suggesting that the authors solve the structure from the broad peaks at higher temperatures, but to simply index them, preferably using a LeBail fit, to see if it is consistent with a primitive tetragonal system.

3.) What is the composition of x in the single crystals? Is it the same for the $T_c = 28$ K sample

included in the Supp Info?

4.) In the papers by Sun et al (Inorg. Chem., 2015, 54, 1958-1964), and Zhou et al (J. Mater. Chem. C, 2016, 4, 3934), there seems to be a correlation between the lattice parameters of $(\text{Li}_{1-x}\text{FexOH})\text{FeSe}$ and the amount of charge doping, and hence T_c . Zhou et al. alleged that the tetragonality ratio (c/a) was a good parameter against T_c . Do the authors find the same behavior with their pressure studies?

5.) I think the letter n is used in two different ways in Figure 1. This gets to be a bit confusing. The authors should redo Fig 1. so that n_e is not confused with the exponent n .

6.) The carrier concentration is increased in the second superconducting dome with respect to the first one. If there is no possible way to further charge dope during the pressure experiment, then how does this occur? The DOS plots in the Supp Info and the Fermi surface figures do not show a major change for the electron pockets. Again, I do believe that there is something going on structurally at higher pressures and doing a bit more analysis with the synchrotron X-ray powder diffraction data could yield the responsible phase. I understand this is a fast-moving competitive field, but it is very important to get it right the first time for the community to make progress.

7.) Finally, did the authors take an X-ray of their sample after the pressure experiment? Does the compound actually survive the pressure experiment and return to the original $(\text{Li}_{1-x}\text{FexOH})\text{FeSe}$ sample?

Our reply to the Referees' report

Reviewer #1:

This manuscript reports the magnetotransport measurements under high pressure up to 12.5 GPa on the single crystal $(\text{Li}_{1-x}\text{Fe}_x)\text{OHFe}_{1-y}\text{Se}$, which is considered to be free from phase separation and relatively stable in air. The authors found that the ambient-pressure SC-I phase is suppressed with increasing pressure to $P_c \sim 5$ GPa, above which a new SC-II phase emerges with T_c increasing with pressure to achieve T_c over 50 K. The reemergence of higher T_c SC-II phase was found to accompany with a concurrent enhancement of electron carrier density. In addition, the authors noted based on the normal state resistivity data a transition from a Fermi liquid for SC-I phase to a non-Fermi-liquid for SC-II phase. The authors concluded that this work provides positive correlations between the high- T_c superconductivity in SC-II with a Fermi surface reconstruction, which is not induced by a structural transition as confirmed by our high-pressure structural study.

This is indeed an interesting new result that is suitable for publication on Nature Communication. However, there are questions the authors need to address before the paper can be accepted for publication:

Our reply: Thank you so much for your careful reading and for your recommendation. We have revised the manuscript according to your suggestions below.

1. The x-ray results—need to show the refinement results to confirm no structural change above 6.29 GPa; the refined lattice parameters that show little changes above 8 GPa, seem not consistent with the XRD patterns.

Our reply: Thanks for your advice. Rietveld structural refinements on the high-pressure SXRD data are quite difficult due to the presence of light elements H, O, and Li, the preferred orientation and peak broadening under high pressures. We thus adopted the LeBail method to extract the lattice parameters only and to examine whether there is structural change or not. The LeBail fitting results given in Fig. 4(a) of the revised manuscript confirm the absence of any structural transition up to at least 10 GPa. It should be noted that the significant peak broadening results in a large uncertainty for the lattice parameters above 10 GPa even with the LeBail method.

2. The appearance of the SC-II above 6 GPa and reaches $T_c \sim 40$ K at ~ 8 GPa is similar to that observed in FeSe under pressure (See Medvedev, et al., DOI:10.1038/NMAT2491). The authors need to provide clear statement to make the distinction between the two.

Our reply: Thanks for your suggestion. The pressure-induced SC-II phase in $(\text{Li,Fe})\text{OHFeSe}$ is distinct from the high- T_c phase of FeSe under high pressure. On the one hand, their maximum T_c values are different. On the other hand, the dominant charge carriers indicated by the slope of Hall resistivity $\rho_{xy}(H)$ in the normal state just

above T_c are opposite: it is negative (electron dominated) for (Li,Fe)OHFeSe, but positive (hole dominated) for FeSe under high pressure (Sun, *et al.*, PRL 118, 147004, 2017). We have pointed these out in the revised manuscript to clearly distinguish the SC-II of LiFeOHFeSe from the high- T_c phase of FeSe under pressure.

Reviewer #2:

The paper by J.-G. Cheng on the reemergence of superconductivity of $(Li_{1-x}Fe_xOH)FeSe$ is quite novel and important for the superconducting community. The conclusions are original and most of the measurements are convincing with respect to finding a second superconducting dome with an even higher T_c in bulk samples. I do not doubt the resistivity or magnetization measurements that ultimately build the phase diagram of Figure 1, but I do have concerns in identifying the phase responsible for this second dome.

Our reply: Thank you so much for your careful reading and for your suggestions.

I suggest the following changes before publication.

1.) The authors state that Rietveld refinements were performed to extract the lattice parameters that are plotted in Figure 4b,c,d. However, the curves in Figure 4a are not fit to anything. I could not find any in the Supplemental Information file either. Somehow, I do not believe that full structural Rietveld fits were performed because the powder patterns do not appear to be of high enough quality to extract lattice parameters, structural parameters (not every atom is in a special position), and other powder parameters (such as background and profile shape). I would suggest that the authors perform, instead, simple LeBail fits and present it as such for the system of interest. This is sufficient for lattice parameters, anyway, and preferred orientation is not even a problem with this type of analysis.

Our reply: Thank you for your advice. Indeed, the Rietveld structural refinements on the high-pressure SXR data are quite difficult with a large uncertainty due to the following factors: the presence of light elements H, O, and Li, the preferred orientation and significant peak broadening under high pressures. We have adopted your suggestion to perform the LeBail fit (The Pattern Match function in the FullProf program) so as to extract the lattice parameters only. The LeBail fitting results are given in Fig. 4 of the revised manuscript. It should be noted that the significant peak broadening still gives a large uncertainty for the lattice parameters above 10 GPa even with the LeBail method.

2.) Not every peak is indexed by the $P4/nmm$ structure in the powder patterns. There is one labeled by an asterisk. What is it? Also what are the extra phases in the pattern? This could be important for the phase that is responsible for the high T_c of 52 K. We are assuming that it is $(Li_{1-x}Fe_xOH)FeSe$, but it might not be. Perhaps it is a new body-centered phase where a collapse of the c -parameter occurs (as in some 122-iron arsenides) that would lead to a complete rearrangement of the electronic structure. Note, that I am not suggesting that the authors solve the structure from the broad

peaks at higher temperatures, but to simply index them, preferably using a LeBail fit, to see if it is consistent with a primitive tetragonal system.

Our reply: We finally figure out that the extra peak around 12° actually comes from the main peak of Selenium (Se) in the space group $P3_121$. As shown in Fig. 4(a) of the revised manuscript, all the peaks in the SXRD pattern can be described excellently up to the highest pressure by including the Se as a secondary phase in the LeBail fit. These results thus confirm the absence of structural transition to a collapsed tetragonal phase up to 14 GPa.

3.) What is the composition of x in the single crystals? Is it the same for the $T_c = 28$ K sample included in the Supp Info?

Our reply: The $(\text{Li}_{1-x}\text{Fe}_x)\text{OHFeSe}$ single crystals studied in the main text are from the same batch as those reported in the previous work, PRB 92, 064515(2015); the x value was determined to be ~ 0.16 based on the structural refinements and ICP-AES analysis. According to our unpublished data and that from literature, e.g. Sun *et al.* Inorg. Chem. 54, 1958 (2015), the sample with a lower $T_c = 28$ K has a similar x value. Instead, the different T_c seems to be related with the lattice constant c as demonstrated in JACS 137 66, (2015). We have added the information on the x value in the revised manuscript.

4.) In the papers by Sun et al (Inorg. Chem., 2015, 54, 1958-1964), and Zhou et al (J. Mater. Chem. C, 2016, 4, 3934), there seems to be a correlation between the lattice parameters of $(\text{Li}_{1-x}\text{Fe}_x\text{OH})\text{FeSe}$ and the amount of charge doping, and hence T_c . Zhou et al. alleged that the tetragonality ratio (c/a) was a good parameter against T_c . Do the authors find the same behavior with their pressure studies?

Our reply: Thanks for pointing out these references. According to the work by Sun *et al.* and Zhou *et al.*, the T_c values of $(\text{Li,Fe})\text{OHFeSe}$ samples increase with increasing the c/a ratio above the critical values around 2.40-2.43. Since the c axis has a larger compressibility for the layered $(\text{Li,Fe})\text{OHFeSe}$, the c/a ratio decreases monotonically at least up to 10 GPa as shown in Fig. 4(c). At a first glance, the initial decreases of T_c up to 5 GPa in the SC-I phase seems to be consistent with the reduction of c/a ratio. However, the value of c/a ratio for $P > 0.8$ GPa is already lower than 2.40, Fig. 4(c), which is out of the superconducting regime for $(\text{Li,Fe})\text{OHFeSe}$ in the above-mentioned references. In addition, T_c of the SC-II phase above 5 GPa increases with pressure despite of the continuous reduction of c/a ratio. Based on these factors, we therefore tend to believe that the T_c of $(\text{Li,Fe})\text{OHFeSe}$ under high pressure does not have a similar connection with the c/a ratio as that found at ambient pressure. We have added sine discussion on this point in the revised manuscript.

5.) I think the letter n is used in two different ways in Figure 1. This gets to be a bit confusing. The authors should redo Fig 1. so that n_e is not confused with the exponent n .

Our reply: Thanks for your advice. We have changed the resistivity exponent n to α in order to avoid any confusion with the electron carrier density n_e . We also made

corresponding changes in the whole manuscript.

6.) *The carrier concentration is increased in the second superconducting dome with respect to the first one. If there is no possible way to further charge dope during the pressure experiment, then how does this occur? The DOS plots in the Supp Info and the Fermi surface figures do not show a major change for the electron pockets. Again, I do believe that there is something going on structurally at higher pressures and doing a bit more analysis with the synchrotron X-ray powder diffraction data could yield the responsible phase. I understand this is a fast-moving competitive field, but it is very important to get it right the first time for the community to make progress.*

Our reply: Thanks for your suggestion. We agree that the structure response is very important for understanding the observed peculiar behaviors under high pressure. We thus have tried our best and also consulted the experienced researchers on the analysis of high-pressure XRD data. Unfortunately, we are still unable to resolve the possible structural transition at higher pressures based on the currently available data. It is also very difficult for us to access the high-pressure synchrotron XRD experiment with the gas pressure transmitting medium in a short period of time. This has to be left for future studies.

Although the application of high pressure does not introduce extra charge carriers, the effective carrier density near the Fermi level can be enhanced via a Fermi surface reconstruction process. According to a recent ARPES study on FeSe films by Phan *et al.* PRB 95, 224507 (2017), a compression strain realized in FeSe/CaF₂ will enlarge significantly both the hole and electron Fermi surfaces in comparison with the strain-free FeSe. It is likely that compression on the FeSe planes above the critical pressure P_c can result in a similar Fermi surface reconstruction or even Lifshitz transition leading to a larger Fermi surface volume. We have added in the revised manuscript some discussion on the possible way for the enlarged Fermi surfaces under high pressure.

7.) *Finally, did the authors take an X-ray of their sample after the pressure experiment? Does the compound actually survive the pressure experiment and return to the original (Li_{1-x}Fe_xOH)FeSe sample?*

Our reply: We are unable to perform a lab XRD measurement on the sample after high-pressure transport measurements because the sample size is quite small, $\sim 0.5 \times 0.3 \times 0.05$ mm³. But we are sure that the sample can keep intact in our high-pressure experiments with the cubic anvil cell apparatus because the sample is immersed in the liquid pressure transmitting medium, as shown below. A visible inspection on the (Li,Fe)OHFeSe sample found no obvious change after releasing pressure from 12.5 GPa. We then performed resistivity measurement on the recovered sample at ambient pressure again and observed a nearly identical resistivity curve with the same T_c as that before loading into the high-pressure cell. The data are shown below.

Reviewer #1 (Remarks to the Author):

The authors have provided sufficient statements to support their views in response to my earlier concerns. The manuscript has also been revised accordingly. Thus, I recommend accepting the revised version for publication in Nature Communication.

Reviewer #2 (Remarks to the Author):

The authors have taken the comments, questions, and suggestions of the reviewers seriously. I appreciate that they now have sought to better explain the structure of the high-pressure phase within the limitations of what can actually be done with such powder data. I appreciate the difficulty of these high-pressure experiments, and really only wanted to see the authors clarify the various powder pattern peaks and identify impurities. I believe this is an important finding for the area of FeSe superconductors and should be published.